# An Initial Survey on Occurrence, Fate, and Environmental Risk Assessment of Organophosphate Flame Retardants in Romanian Waterways

Iuliana Paun [†], Florinela Pirvu [†], Vasile Ion Iancu [ID], Marcela Niculescu, Luoana Florentina Pascu * and Florentina Laura Chiriac *[ID]

National Research and Development Institute for Industrial Ecology—ECOIND, Drumul Podu Dambovitei Street 57-73, 060652 Bucharest, Romania; iuliana.paun@incdecoind.ro (I.P.); florinela.pirvu@incdecoind.ro (F.P.); vasile.iancu@incdecoind.ro (V.I.I.); marcela.niculescu@incdecoind.ro (M.N.)

* Correspondence: ecoind@incdecoind.ro (L.F.P.); laura.chiriac@incdecoind.ro (F.L.C.)
† These authors contributed equally to this work.

**Abstract:** Organophosphate ester flame retardants (OPFRs) are ubiquitous organic pollutants in the environment and present an important preoccupation due to their potential toxicity to humans and biota. They can be found in various sources, including consumer products, building materials, transportation industry, electronic devices, textiles and clothing, and recycling and waste management. This paper presents the first survey of its kind in Romania, investigating the composition, distribution, possible sources, and environmental risks of OPFRs in five wastewater treatment plants (WWTPs) and the rivers receiving their effluents. Samples from WWTPs and surface waters were collected and subjected to extraction processes to determine the OPFRs using liquid chromatography with mass spectrometric detection. All the target OPFRs were found in all the matrices, with the average concentrations ranging from 0.6 to 1422 ng/L in wastewater, 0.88 to 1851 ng/g dry weight (d.w.) in sewage sludge, and 0.73 to 1036 ng/L in surface waters. The dominant compound in all the cases was tri(2-chloroisopropyl) phosphate (TCPP). This study observed that the wastewater treatment process was inefficient, with removal efficiencies below 50% for all five WWTPs. The environmental risk assessment indicated that almost all the targeted OPFRs pose a low risk, while TDCPP, TCPP, and TMPP could pose a moderate risk to certain aquatic species. These findings provide valuable information for international pollution research and enable the development of pollution control strategies.

**Keywords:** OPFRs; LC-MS/MS; wastewater; sewage sludge; surface water; environmental risk

## 1. Introduction

With rapid social development, freshwater ecosystems have become increasingly vulnerable to severe phosphorous pollution [1]. Phosphorous is extensively used in the production of fire retardants, electronics, pesticides, polyurethane foams, fertilizers, and textiles [2,3]. Consequently, the amount of phosphorous in the freshwater environment has increased due to uncontrolled fertilizer and pesticide use, as well as the discharge of domestic and industrial wastewater in natural water body receivers [4]. Studies have shown that a total phosphorous concentration exceeding 0.02 mg/L in a freshwater environment can lead to eutrophication, posing a potential threat to aquatic ecosystems [5]. Therefore, it is essential to control and monitor the level of organophosphorus substances in the aquatic environment to ensure the safety of freshwater ecosystems. Organophosphorus substances constitute a significant fraction of the total phosphorous in the aquatic medium. In recent years, numerous organophosphorus chemicals have entered human communities due to the rapid advancement of phosphorous chemical manufacturing. One notable class of organophosphorus compounds is organophosphorus flame retardants (OPFRs),

which are used in the production of polyurethane foams, building materials, textiles, and coatings. These compounds have emerged as environmental pollutants and a significant environmental concern. It has been estimated that their usage will reach up to 7.5 billion pounds by 2027, with an annual growth rate of 4.4% between 2022 and 2027 [6]. As OPFRs are considered alternatives to halogenated flame retardants, their utilization now accounts for 20% of the total application of flame retardants in Europe [7].

The various negative effects of OPFRs on human health have been reported in the literature, including dermatitis, eye/ear inflammation, carcinogenicity, neurotoxicity, and damaged physical evolution [7–9]. These negative consequences on both the environment and human health have led to global regulations on OPFRs [10]. Due to the daily exposure of humans to OPFRs, special attention has been focused on the group of flame-retardant chlorinated phosphate esters, namely Tris(chloroethyl)phosphate (TCEP), Tris(2-chloroisopropyl) phosphate (TCPP) and Tris(1,3-dichloro-2-propyl) phosphate (TDCPP). The Danish Environmental Protection Agency has expressed concern about their presence in products intended for use by children [11]. As a result, the European Commission has established maximum allowed limit values for the presence of these substances in toys, which is set at 5 mg/kg [12]. Recently, several environmental organizations have submitted a petition to the US EPA requesting additional tests for TCEP, TCPP, and TDCPP. These tests aim to assess the potential risk these substances pose to both human health and the environment in terms of toxicity and persistence [13].

OPFRs are registered as large volume manufacturing chemicals and have become ubiquitous in the environment, present in various sources such as wastewater [3,14], surface water [14,15], groundwater [16], seas/oceans [17–19], sludge [20], sediment [21], soil [22], to drinking water [14,23], household dust [24], humans [25,26], and other aquatic and terrestrial organisms [27,28]. Numerous studies have potential health issues associated with the presence of OPFRs. For example, tributyl-phosphate (TBP) has been linked to sick house syndrome, while tris(chloroethyl) phosphate (TCEP), Tris(1,3-dichloro-2-propyl) phosphate (TDCCP), tris(2-chloroisopropyl) phosphate (TCPP) and tris(2-butoxyethyl) phosphate (TBEP) are considered carcinogenic. Neurotoxic effects have been associated with tripropyl phosphate (TPP) and TBP, while hormone disruption and a decrease in semen quality have been observed with TPP and TDCPP [29,30].

Different analytical procedures and extraction methods have been developed for the detection of OPFRs in the aqueous environment. These methods are often coupled with chromatographic techniques such as gas chromatography or liquid chromatography with mass spectrometric detection (GC-MS and LC-M) [3,31]. While gas chromatography has been commonly used for detecting OPFR compounds in different environmental compartments, an increasing number of LC-MS/MS methods have been developed in recent years [32,33]. Among the main reasons why more and more researchers are using the liquid chromatographic technique, it is worth mentioning the interferences encountered when using gas chromatography with an electron impact ionization source. These interferences can include a lack of structural information in the mass spectra of some OPFRs, false positive results, and co-elution due to the presence of compounds containing phosphorus in the analyzed matrix [34,35]. In comparison, the use of the electrospray ionization source in LC-MS/MS techniques improves their selectivity. Additionally, high specificity can be achieved using multiple reaction monitoring (MRM). The LC-MS/MS technique is also suitable for the detection of mono- and diesters of phosphoric acid, which are not sufficiently volatile for GC techniques [36].

Romania is endowed with diverse freshwater ecosystems, including rivers, lakes, wetlands, and groundwater reserves. These ecosystems support a rich array of plant and animal species and provide numerous ecosystem services. However, they face a multitude of threats, particularly pollution originating from various sources such as agricultural runoff, industrial discharges, and inadequate wastewater treatment systems. The extent of pollution in Romanian freshwater ecosystems has becomes a pressing concern. Nutrient pollution, caused by the excessive use of fertilizers in agriculture and ineffective waste

management practices, has led to eutrophication in many water bodies. Additionally, heavy metals, pesticides, and organic pollutants from industrial activities have further deteriorated the water quality and disrupted aquatic ecosystems. In addition to addressing these general challenges, the region faces unique environmental issues. For instance, the Danube River, which flows through Romania, is one of the most international rivers in the world, posing challenges for cross-border cooperation in managing its water resources. Furthermore, the development of hydropower projects in the Carpathian Mountains has raised concerns regarding their potential impact on freshwater ecosystems and migratory fish populations.

Although OPFR compounds have been widely studied in Europe, no environmental monitoring has been conducted in Romania until now [37,38]. Aquatic systems play a significant role in the migration of OPFRs in the environment. Surface water and ground-water are important environmental compartments for the ecosystem and are often used as drinking water sources. Additionally, the release of wastewater has been recognized as a potential source of contamination of OPFR compounds in aquatic ecosystems [14]. Therefore, it is extremely important to evaluate the concentration levels of OPFRs in aquatic environments for the health of both wildlife and humans.

Despite the gradual phase-out of OPFRs, their potential risks should not be overlooked. Therefore, this paper aimed to analyze eleven OPFR compounds in the wastewater and sewage sludge of five wastewater treatment plants (WWTPs) in Romania. The study also assessed the distribution, mass loading, mass emission and removal of OPFRs in the five WWTPs based on the determined values in the samples. Furthermore, the concentration of OPFRs in natural freshwater bodies receiving effluent discharge was determined. The ecological risks of the targeted OPFRs in the effluents and rivers were evaluated using the risk quotient (RQ) method. To date, no information regarding the occurrence of OPFRs in the Romanian environment is available.

In our study, we focused on analyzing OPFRs from water matrices (surface water and wastewater) and sewage sludge using a combination of extraction techniques, namely, solid-phase extraction (SPE) for water samples and ultrasound-assisted extraction (UAE) for sewage sludge. Liquid chromatography-tandem mass spectrometry (LC-MS/MS) was then employed as the monitoring technique to detect and quantify OPFRs in the extracted samples. SPE is a widely used technique for sample preparation in environmental analysis. The water samples are passed through an SPE cartridge containing the sorbent material, allowing for the retention of OPFRs while other unwanted compounds are washed away. The retained OPFRs are then eluted from the cartridge with a suitable solvent for subsequent analysis by LC-MS/MS. UAE involves subjecting the sludge sample to ultrasonic waves, which create cavitation and enhance the extraction efficiency. The application of ultrasound not only improves the release of target analytes from the sludge matrix but also reduces the extraction time compared to traditional extraction methods. By utilizing UAE, we aimed to achieve the maximum extraction of OPFRs from the sewage sludge samples, facilitating their subsequent analysis. LC-MS/MS, as the monitoring technique, demonstrated excellent sensitivity and selectivity for the detection and quantification of OPFRs. The method allowed for the reliable identification and accurate quantification of OPFRs, even at trace levels, in complex environmental matrices. By combining the extraction methods with LC-MS/MS analysis, we were able to obtain comprehensive and reliable data on the occurrence and concentration of OPFRs in water matrices and sewage sludge. The chosen techniques ensured the efficient extraction, purification, and detection of OPFRs, providing valuable insights into their presence and distribution in the environment.

## 2. Materials and Methods

### 2.1. Chemicals and Reagents

Analytical purity standards, Tri(2-chloroethyl) phosphate (TCEP), tripropyl phosphate (TPP), tri(2-chloroisopropyl) phosphate (TCPP), tri(1,3-dichloro-2-propyl) phosphate (TDCPP), tri(1,3-dibromopropyl)phosphate (TDBPP), tri(2-eylhexyl)phosphate (TMPP),

triphenyl phosphate (TPHP) and the metabolites dibutyl phosphate (DBP), bi(2-ethylhexyl) phosphate (BEHP), and diphenyl phosphate (DPHP) were obtained from Sigma-Aldrich (Darmstadt, Germany). The physical–chemical parameters of the targeted compounds are provided in Table S2. The solvents used for the mobile phase preparation and extraction from water and sludge samples were acetonitrile and methanol from Merck (Darmstadt, Germany), and mobile phase modifier formic acid (HCOOH) was obtained from Sigma-Aldrich (Darmstadt, Germany). Stata-X Polymeric Reverse Phase SPE cartridges (33 μm, 500 mg/6 mL) were purchased from Phenomenex, and silica gel was supplied by Sigma Aldrich (Darmstadt, Germany).

## 2.2. Sampling Design

In this study, a carefully designed sampling approach was employed to capture the presence and transport of organophosphate flame retardants (OPFRs) across the studied environments, including influents, effluents, sewage sludge, and receiving rivers.

Influent sampling was conducted at the entry point of the wastewater treatment plants to assess the input of OPFRs into the system. The influent samples represented a mixture of domestic and industrial wastewater, serving as a representative source of OPFR contamination. By collecting influent samples, we aimed to determine the initial concentration of OPFRs entering the wastewater treatment plants.

Effluent sampling was performed to evaluate the effectiveness of the wastewater treatment processes in removing OPFRs. The treated effluents from the wastewater treatment plants were sampled to determine the concentration of OPFRs that remained after treatment. This analysis allowed us to quantify the extent to which OPFRs were eliminated during conventional wastewater treatment and to identify any potential releases of these compounds into the environment through treated effluents.

Sampling of sewage sludge was conducted as it represents the solid residue produced during the wastewater treatment process. Sewage sludge samples were collected to assess the accumulation of OPFRs in the sludge. This analysis helped determine the potential for OPFRs to be transported further through land application or disposal of the sludge, leading to their migration into soil and nearby water bodies.

Sampling of receiving rivers was carried out to investigate the downstream transport and dispersion of OPFRs from the wastewater treatment plants. By analyzing these samples, we aimed to determine the extent of OPFR contamination in the receiving rivers and to identify any potential ecological risks associated with their presence.

By utilizing this comprehensive sampling design, we aimed to gain a thorough understanding of the primary pathways along which OPFRs are transported across the studied environments. This would provide valuable insights for assessing the environmental fate and potential impacts of these compounds.

## 2.3. Sample Collection

Composite samples of influent, effluent, and sludge were collected from five WWTPs in Romania, in 2022. The samples were collected over a 24 h period. These WWTPs receive municipal wastewater from five important cities. Surface water samples were taken from approximately 50 m upstream and downstream of the effluent discharge areas on the rivers Ialomita, Siret, Dambovita, Bahlui, and Olt, during the same time period as the wastewater samples. The WWPTs' characteristics and sample codification can be found in Table S1. All samples were collected in glass containers. A total of 15 samples from each matrix (five samples from each matrix collected in triplicate), including water matrices and sewage sludge, were collected. The samples were kept at 4 °C during transport and later stored at −20 °C until analysis.

## 2.4. Sample Preparation

The extraction of OPFR compounds from water samples was conducted using the solid phase extraction (SPE) procedure, with an automatic solid phase extraction system,

SPE AutoTrace 280 Thermo Scientific Dionex. Prior to extraction, the water samples (200 mL surface water and 100 mL wastewater) were filtered and spiked with 1 mL of internal standard solution (10 µg/L). SPE Strata-X Polymeric Reverse Phase cartridges were conditioned with 10 mL each of acetonitrile and water. After elution of the samples, the cartridges were washed with 10 mL ultrapure water, and the adsorbent phase was dried for 40 min under a nitrogen stream. The analytes were subsequently eluted from the SPE cartridges using two consecutive 10 mL portions of acetonitrile, which were collected in concentration tubes. The extracts were evaporated almost to dryness and resuspended using 1 mL of ultrapure water.

The extraction of OPFR compounds from sludge samples was performed using the ultrasound-assisted extraction procedure. The sludge samples were first lyophilized at −110 °C and then homogenized. A weight of 0.5 g of the homogeneous mixture was measured and placed into 20 mL ampoules. These samples were then spiked with 1 mL of a 10 µg/L internal standard solution. Afterwards, 10 mL of acetonitrile was added to the samples, which were then subjected to ultrasonication for 15 min followed by centrifugation for 10 min at 4000 RPM. The extraction process was repeated one more time using 10 mL of acetonitrile. The supernatants were combined and evaporated to near dryness in a water bath at 40 °C, using a gentle stream of nitrogen. The resulting samples were then redissolved in 1.0 mL water and transferred to vials for analysis.

### 2.5. Instrumental Analysis

An Agilent 1260 liquid chromatography system linked with an Agilent 6410B tandem mass spectroscopy device (Agilent Technologies, Santa Clara, CA, USA) was employed for the determination of OPFRs. Separation of OPFRs was achieved using a Zorbax Eclipse Plus C18 chromatographic column (150 × 2.1 mm, 3.5 µm, Agilent). A binary mobile phase consisting of 0.1% formic acid in water and 0.1% formic acid in methanol was used in a gradient, and the target OPFRs were analyzed using the multiple reaction monitoring (MRM) mode. More information on the analytical method's conditions can be found in the Supplementary Materials (Tables S3–S6). Both negative and positive modes were employed using an electrospray ionization source (ESI). The ESI parameters were as follows: drying gas flow—8 L/min, capillary voltage—4000 V, nebulizer pressure—40 psi. A specific MRM chromatogram is illustrated in Figure S1.

### 2.6. Quality Assurance and Quality Control

The calibration curves were fitted using seven calibration levels, ranging from 0.1 to 200 µg/L, with correlation coefficients ($R^2$) higher than 0.99 for all compounds. All experiments were conducted in triplicate. To prevent potential sample contamination, all laboratory articles used to investigate the OPFR were pre-cleaned with CAN. Field blanks and procedural blanks were analyzed at every batch. Results from the blank samples were below the method's detection limits (LODs) for all analytes. For intra-day precision, the relative standard deviation (RSD) ranged from 3.83% to 6.65% for surface water, 4.23% to 6.83% for wastewater, and 5.23% to 7.53% for sludge. The inter-day precision showed RSD values ranging from 6.16 to 9.94% for surface water, 8.25% to 11.3% for wastewater, and 10.1% to 12.6% for sludge (Table S7). The recoveries of the OPFR compounds, assessed by spiking the 11 analytes in all matrices, ranged from 75% to 114% (Table S8), depending on the sample types. The matrix effects evaluation showed results ranging from 74% to 115% for surface water, 70% to 101% for influents, from 71% to 105% for effluent samples, and from 73% to 121% for sludge samples (Table S9). The LODs were determined as the lowest concentration at which the signal to noise ratio (S/N) was equal to 3, while the limits of quantitation (LOQs) were determined as the lowest concentration with S/N = 10. LOQ values obtained were between 0.36 ng/L and 0.95 ng/L for surface water, 1.42 ng/L and 1.04 ng/L for wastewater, and 0.16 ng/g d.w. and 0.67 ng/g d.w. for sludge (Table S10).

*2.7. Results Evaluation*

The removal efficiencies (REs) of the targeted OPFRs during the treatment processes in WWTPs were evaluated using Equation (1):

$$RE = \frac{(Cinf - Cef)}{Cinf} \times 100 \tag{1}$$

where Cinf is the OPFRs concentration in influent samples and Cef is the OPFRs concentration in effluent sample.

The daily mass loading (DML) and mass emission (DME) (mg/day/1000 persons) of OPFR compounds was calculated for each influent and effluent, respectively, using Equations (2) and (3) [39]:

$$DML = (Q \times Cinf)/P \tag{2}$$

$$DME = (Q \times Cefl)/P \tag{3}$$

where Q is the daily flow rate of the treatment plant ($m^3$/day), and P is the population served by the WWTPs.

Environmental risk posed by the targeted OPFRs for aquatic organisms was evaluated using the risk quotient (RQ). The RQs were calculated using the following equation:

$$PNEC = NOEC\ (LC_{50}\ sau\ EC_{50})/AF \tag{4}$$

$$RQ = MEC/PNEC \tag{5}$$

where PNEC represents the predicted no-effect concentration, NOEC refers to the no-effect concentration, MEC denotes the OPFRs concentration determined in water samples, $L(E)C_{50}$ represents the toxicity information provided from literature, and AF represents the assessment factor. The AF value was determined in accordance with the technical guidance document (TGD) on risk assessment of the European Commission: 1000 applies for $LC_{50}/EC_{50}$; 100 applies for a long-term NOEC; 50 and 10 apply to two and three long-term NOECs in species representing different trophic levels, respectively [17,40,41].

Usually, the common criterion for risk level is performed by interpreting RQs as low risk (RQ < 0.1), medium risk (0.1 < RQ < 1), and high risk (RQ > 1) [17].

*2.8. Statistical Analysis*

Statistical analysis, including the calculation of Relative Standard Deviation (RSD), minimum, maximum, average values, and detection frequency, was performed using Microsoft Excel software. Excel provides various functions and tools that allow for data analysis and calculation of statistical parameters. These calculations can be performed using formulas and built-in functions such as AVERAGE, MIN, MAX, STDEV, and COUNTIF, among others. To identify any significant differences ($p \leq 0.05$) among the five chosen WWTPs, we conducted the two-way ANOVA test in the EXCEL program, using the data analysis function.

## 3. Results and Discussion

*3.1. Occurrence of OPFRs in WWTPs*

3.1.1. Occurrence and Concentration of OPFRs in Wastewater

All the target compounds were determined in the wastewater samples, with the exception of only TDBPP, for which the determined frequency was only 20% (Table 1). The combination of the variability in the discharge, dilution and mixing, transformation, and degradation processes, as well as sorption and binding, can contribute to the observed determined frequency of 20% for TDBPP in both the WWTP influent and effluent samples. Further research and sampling from a larger number of WWTPs or over an extended period may help provide a more comprehensive understanding of TDBPP occurrence and fate during wastewater treatment processes. In the influent samples, TCPP was

found to be the predominant compound, with an average concentration ranging from 1301 ng/L to 1711 ng/L (Figure 1). Similar values have been reported in the literature, with up to 1244 ng/L in the USA [42], up to 2500 ng/L in Sweden [43], up to 2400 ng/L in Germany [15], up to 3872 ng/L in China [44], and up to 2500 ng/L in Australia [45]. Lower values have been reported in Spain (up to 410 ng/L) [46] and Canada (up to 380 ng/L) [47]. The second highest compound, in terms of the concentration, after TCPP, was the DBP metabolite, with a concentration ranging from 39 ng/L to 173 ng/L. The concentrations of the other compounds were in the tens of ng/L range. Higher concentrations have been reported in studies conducted in EU countries such as Germany [15] and Sweden [44], as well as in the USA [42]. The order of the decreasing concentrations of the OPFRs in the influent samples is as follows: TCPP > DBP > TDCPP > TEHP > DPHP > BEHP = TCEP > TDBPP = TMPP > TPHP > TPP. Table 1 presents the minimum and maximum values, average values, as well as the frequency of determination, for both the influent and effluent samples.

**Table 1.** Minimum, maximum, and average concentrations (ng/L) and determination frequencies (%) of OPFRs in the wastewaters.

| OPFRs | Influent | | | | Effluent | | | |
|---|---|---|---|---|---|---|---|---|
| | **Min** | **Max** | **Average** | **Freq.** | **Min** | **Max** | **Average** | **Freq.** |
| DPHP | 5.9 | 34 | 13 | 100 | <LOQ | 8.2 | 2.8 | 100 |
| DBP | 39 | 173 | 85.0 | 100 | 3.7 | 94 | 47 | 100 |
| BEHP | 10 | 26 | 17.6 | 100 | 3.7 | 17 | 10 | 100 |
| TCEP | 11 | 26 | 21 | 100 | 26.8 | 49 | 36 | 100 |
| TPP | <LOQ | 1.5 | 0.9 | 80 | <LOQ | 0.9 | 0.6 | 100 |
| TCPP | 1301 | 1711 | 1422 | 100 | 757 | 1299 | 1006 | 100 |
| TDCPP | 23 | 46 | 35.3 | 100 | 25 | 47 | 36 | 100 |
| TPHP | 2.7 | 11 | 6.7 | 100 | 1.8 | 3.6 | 2.6 | 100 |
| TDBPP | ND | 15 | 3.0 | 20 | ND | 19 | 3.7 | 20 |
| TMPP | 6.0 | 15 | 9.0 | 100 | 1.7 | 6.8 | 4.2 | 100 |
| TEHP | 21 | 35 | 25.1 | 100 | 8.6 | 19 | 13 | 100 |

ND—undetected.

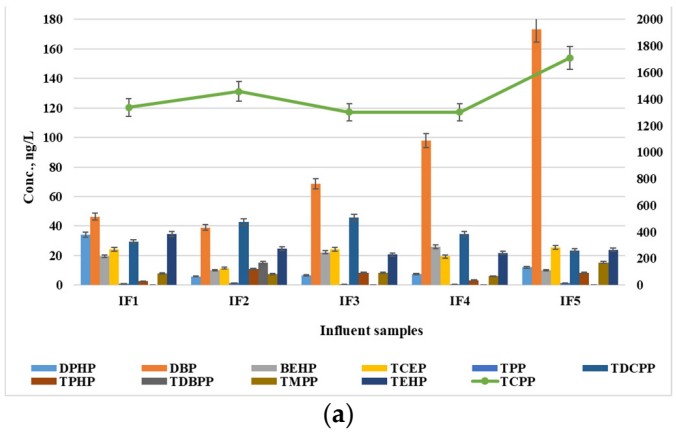 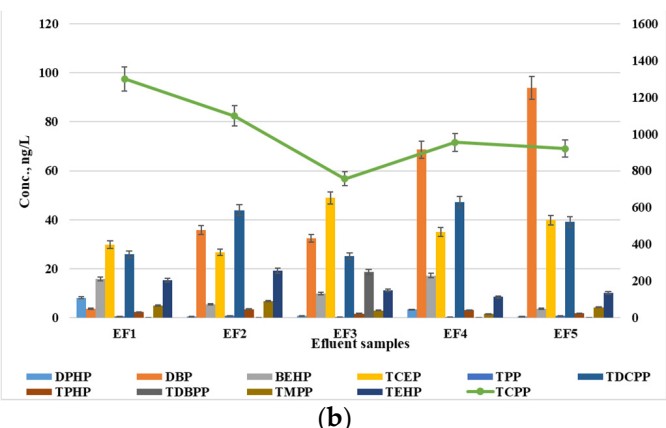

(**a**)        (**b**)

**Figure 1.** Concentration levels of OPFRs in wastewater samples: (**a**) influents and (**b**) effluents.

The results of the two-way ANOVA test suggest that there are no statistically significant differences in the OPFR concentrations among the influent samples collected from the five WWTPs. This finding is important, given that the wastewater treatment steps are identical for all the WWTPs, involving mechanical and biological purification followed by a tertiary stage for nutrient removal. The between-group analysis revealed a non-significant F-value of 0.18 with 4 degrees of freedom, compared to a critical F-value of 3.26. The associated *p*-value of 0.95 indicates that there is a 95% chance of obtaining these results by chance alone, far exceeding the typical significance level of 0.05. Thus, we fail to reject the null

hypothesis, suggesting no statistically significant differences in the OPFR concentrations among the WWTPs. Similarly, the within-group analysis yielded a non-significant F-value of 1.75 with 3 degrees of freedom, compared to a critical F-value of 3.49. The associated *p*-value of 0.22 indicates a 22% chance of obtaining these results by chance alone, which is considerably higher than the typical significance level. Therefore, no statistically significant differences in the OPFR concentrations within the WWTPs were observed.

However, it is crucial to note that although the ANOVA test did not detect significant differences, there might still be some level of variation between the WWTPs that was not detected in this study. Conducting further investigations could provide additional insights into any potential differences not captured in the current analysis.

In the effluent samples, TCPP also showed the highest concentrations, ranging from 757 ng/L to 1299 ng/L (Figure 1). Higher values were determined in Sweden (up to 2000 ng/L) [43], Germany (up to 1700 ng/L) [15], and the USA (up to 4765 ng/L) [42], and were lower in Spain (<500 ng/L) [46], China (<54 ng/L) [40], and Austria (<674 ng/L) [47]. Similar to the influent samples, DBP was the second most abundant compound, with concentrations ranging from 3.7 ng/L to 94 ng/L. TCEP and TDCPP were identified at very similar concentration levels in the effluents, ranging from 25 ng/L to 49 ng/L. Higher concentrations have been reported in similar studies conducted in Sweden [43], Spain [46], Germany [48], Austria [49], and the USA [42].

The statistical analysis results indicate that there are no statistically significant differences in the concentrations of OPFRs among the effluent samples from the five wastewater treatment plants. This conclusion is supported by the *p*-values obtained for both the between-group and within-group analyses, which exceed the significance level of 0.05. The between-group analysis yielded a non-significant F-value of 0.83 with 4 degrees of freedom, compared to a critical F-value of 3.26. This suggests that any differences observed in the OPFR concentrations between the treatment plants are likely due to random chance rather than significant variations. Therefore, we can conclude that there are no significant differences in the OPFR concentrations among the treatment plants. Similarly, the within-group analysis produced a non-significant F-value of 1.57 with 3 degrees of freedom, compared to a critical F-value of 3.49. The associated *p*-value of 0.14 indicates a 14% chance of obtaining these results by chance alone, which is also higher than the typical significance level. This further supports the conclusion that there are no significant differences in OPFR concentrations within the treatment plants. Thus, the statistical analysis suggests that the concentrations of the OPFRs in the effluent samples from the five wastewater treatment plants do not vary significantly. This implies that the wastewater treatment processes employed by each plant are comparably effective in reducing OPFR concentrations.

The information provided in Figure 2 highlights interesting trends in the concentrations of the OPFRs in the influents and effluents of the different wastewater treatment plants based on the population served. Additionally, it mentions the main sources of contamination and the values measured for the OPFRs.

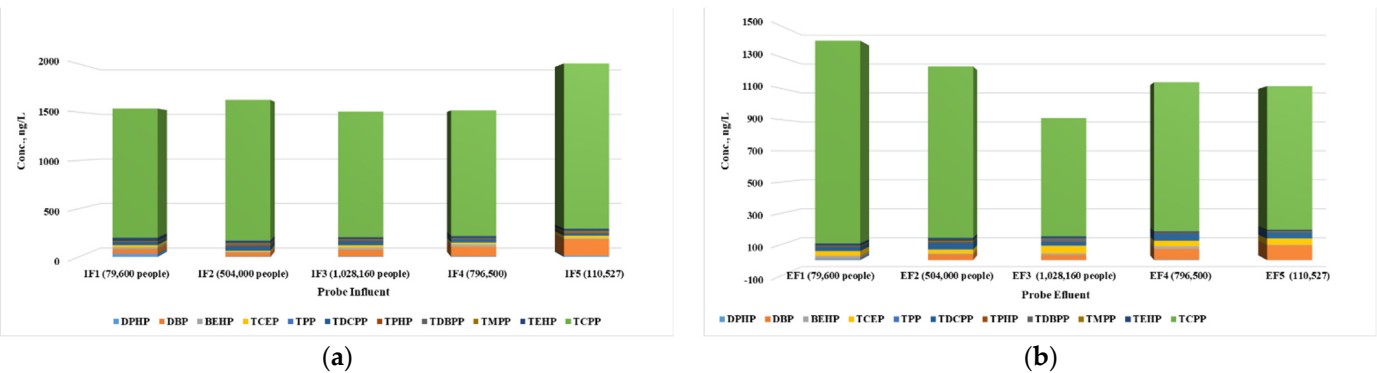

**Figure 2.** Composition of OPFRs in wastewater samples: (**a**) influents and (**b**) effluents.

The data suggest that the treatment plant serving a relatively smaller population (S1) has the highest sum of OPFR concentrations in the influent, while the treatment plant serving a larger population (S3) has the lowest sum. This observation may be attributed to specific pollution sources present in the catchment area of S1, such as industrial activities or other point sources of OPFR contamination. It is important to further investigate and identify the specific sources responsible for the higher influent concentrations in S1. In terms of effluents, S1 exhibits the highest sum of OPFR concentrations, followed by S2, S4, and S5, with S3 recording the smallest value. This suggests that the treatment processes at S1 are less effective in removing OPFRs compared to other treatment plants. It is crucial to understand the reasons behind this inefficiency, such as the treatment technologies employed or the quality of influent received. Improving the treatment processes at S1 to enhance OPFR removal should be considered to minimize environmental impacts.

Interestingly, the treatment plant serving the largest population (S3) has the lowest concentration of OPFRs in the effluent. This signifies that the treatment processes implemented at S3 are successful in effectively reducing the OPFR levels. It would be valuable to investigate the specific treatment methods employed at S3 to identify any contributing factors to this successful removal. When considering the relation between population and contamination sources, it is important to note that the population size alone does not solely determine the concentrations of OPFRs. Other factors like industrial activities, agricultural practices, and proximity to potential pollution sources can significantly influence the contamination levels.

In conclusion, the concentrations of OPFRs in the wastewater treatment plants' influents and effluents showcase variations related to the population, sources of contamination, and treatment efficiency. Further research is needed to identify specific contamination sources and optimize treatment processes to ensure the effective removal of OPFRs and minimize their potential environmental and health risks.

### 3.1.2. Removal Efficiencies of OPFRs in the Five WWTPs

The removal efficiency was calculated for each individual compound as well as the sum of compounds for each WWTP (Figure 3). In all five WWTPs, higher concentrations of TCEP were observed in the effluents compared to the influents. Additionally, higher concentrations of TDCPP were observed in the effluents of the treatment plants S4 and S5 compared to their corresponding influent samples. It is likely that these compounds are formed during the treatment processes. However, the removal of these compounds was generally incomplete, with a removal yield exceeding 70% only in a few cases.

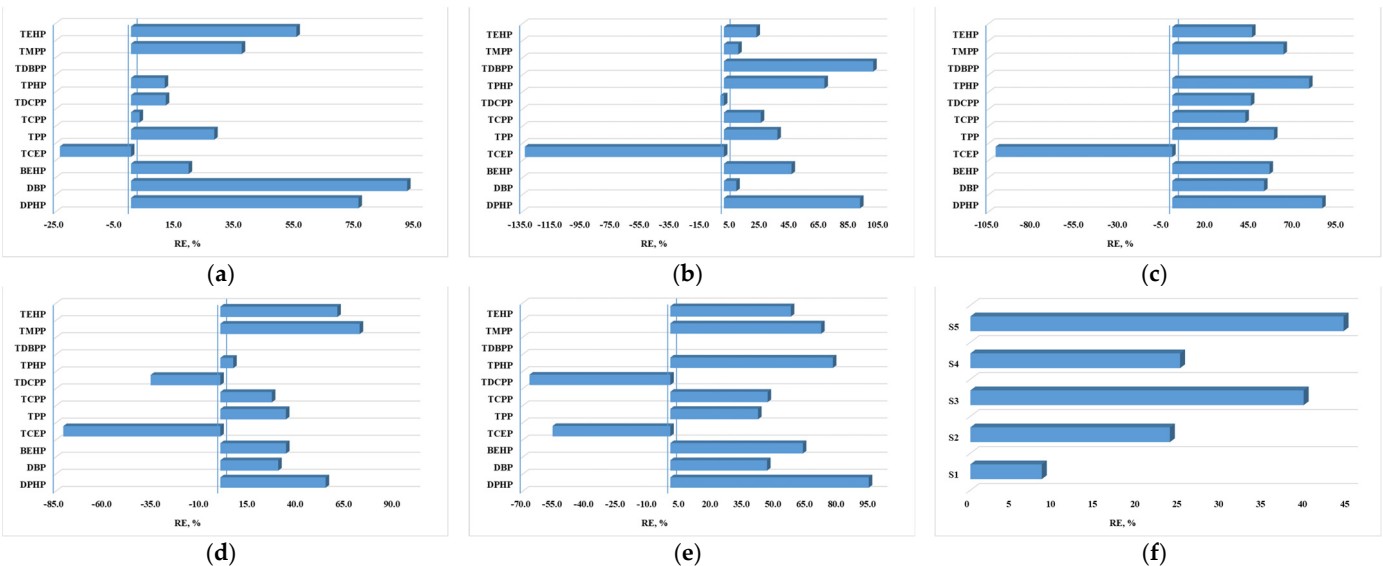

**Figure 3.** The removal efficiencies determined for individual compounds in WWTPs: (**a**) S1; (**b**) S2; (**c**) S3; (**d**) S4; (**e**) S5; and (**f**) sum of OPFRs.

The removal efficiency values calculated based on the sum of the OPFR compounds were below 50% for all the WWTPs, indicating that the treatment steps are not sufficient to completely remove them. This study demonstrates significant differences in the elimination of chlorinated organophosphorus flame retardants compared to the non-chlorinated ones. While chlorinated organophosphates like TCPP, TCEP, and TDCP were not efficiently eliminated during the treatment process, non-chlorinated derivatives such as TPHP, TPP, and organophosphate diesters showed partial elimination. Similar results have been reported in the literature [50,51].

### 3.1.3. Daily Mass Loading and Mass Emission

The daily mass loading (DML) and daily mass emission (DME) levels of the WWTPs are presented in Table 2. The highest DML value was calculated for TCPP (650–1364 mg/day/ 1000 people). High mass loadings were also observed for DBP, with up to 138 mg entering the WWTPs on a daily basis. For the other OPFR compounds, the mass load did not exceed 35 mg/day/1000 people. The emission of OPFRs from WWTPs into surface waters ranged from 0.2 and 935 mg/day/1000 people. The highest DME level was determined for TCPP, the values reaching hundreds of mg/day/1000 people.

**Table 2.** DML and DME values determined for the targeted OPFRs.

| Daily Mass Loading (mg/zi/1000 People) | | | | | | | | | | |
|---|---|---|---|---|---|---|---|---|---|---|
| Samples | DPHP | DBP | BEHP | TCEP | TPP | TCPP | TDCPP | TPHP | TDBPP | TMPP | TEHP |
| IF1 | 20.4 | 27.8 | 11.7 | 14.5 | 0.53 | 800 | 17.6 | 1.64 | - | 4.71 | 20.6 |
| IF2 | 2.62 | 17.4 | 4.52 | 5.14 | 0.65 | 650 | 19.1 | 4.88 | 6.83 | 3.46 | 11.3 |
| IF3 | 3.84 | 138 | 12.5 | 13.7 | 0.44 | 731 | 25.8 | 4.57 | - | 4.68 | 11.6 |
| IF4 | 7.31 | 95.9 | 25.6 | 19.2 | 0.41 | 1275 | 34.2 | 3.33 | - | 5.92 | 21.3 |
| IF5 | 9.65 | 138 | 8.23 | 20.4 | 1.26 | 1364 | 18.7 | 6.62 | - | 12.3 | 19.1 |
| Daily Mass Emission (mg/zi/1000 People) | | | | | | | | | | |
| Samples | DPHP | DBP | BEHP | TCEP | TPP | TCPP | TDCPP | TPHP | TDBPP | TMPP | TEHP |
| EF1 | 4.94 | 2.17 | 9.53 | 17.9 | 0.45 | 777 | 15.6 | 1.46 | - | 2.98 | 9.31 |
| EF2 | 0.25 | 16.2 | 2.36 | 12.4 | 0.38 | 489 | 19.6 | 1.62 | - | 3.11 | 8.64 |
| EF3 | 0.53 | 68.3 | 5.48 | 27.5 | 0.22 | 425 | 14.2 | 1.12 | 10.5 | 1.74 | 6.35 |
| EF4 | 3.34 | 67.2 | 16.9 | 34.4 | 0.31 | 935 | 46.2 | 3.25 | - | 1.62 | 8.37 |
| EF5 | 0.65 | 74.8 | 3.49 | 31.8 | 0.74 | 734 | 31.3 | 1.54 | - | 3.53 | 8.28 |

### 3.1.4. Occurrence and Concentration of OPFRs in Sewage Sludge

All 11 OPFRs were determined in the sludge samples with a frequency of 100%, except for TDCPP, which was only detected in 3 out of 5 samples. The most abundant compound was TCPP, with a concentration ranging from 36 ng/g d.w. to 2178 ng/g d.w. (Table 3, Figure 4a). These concentrations are higher compared to the results reported in China (up to 378 ng/g s.u.) [30] and Canada (up to 196 ng/g d.w.) [47].

TDBPP (up to 100 ng/g d.w.) and BEHP (up to 655 ng/g d.w.) also showed relatively high values. The TCEP concentration ranged from 16 ng/g d.w. to 68 ng/g d.w., which falls within the range reported in Canada (up to 21.5 ng/g d.w.) [47], while China reported higher concentrations (<LOD-208 ng/g d.w.) [52]. The concentration of the other compounds was in the range of a few ng/L (DPHP, DBP, TPP) and several tens of ng/L (TPHP, TMPP, and TEHP).

The statistical analysis performed on the sewage sludge samples indicated that there were no significant differences in the concentrations of the OPFRs among the samples. This conclusion is supported by the *p*-values obtained for both the between-group and within-group analyses, which exceeded the predetermined significance level of 0.05. Specifically, the between-group analysis yielded a non-significant F-value of 1.55 with 4 degrees of freedom, compared to a critical F-value of 3.26 (*p*-value = 0.25). This suggests that any observed differences in the OPFR concentrations between the sewage sludge samples from

different sources are likely due to random variation rather than meaningful distinctions. Therefore, it can be concluded that there are no significant differences in the OPFR concentrations among the sewage sludge samples. Similarly, the within-group analysis produced a non-significant F-value of 794 with 3 degrees of freedom, compared to a critical F-value of 3.49 (*p*-value = 0.92), which supports the conclusion that there are no significant differences in the OPFR concentrations within the sewage sludge samples. Thus, the statistical analysis suggests that there are no meaningful variations in the concentrations of the OPFRs among the sewage sludge samples.

**Table 3.** Minimum, maximum, and average concentrations (ng/g d.w.) and determination frequencies (%) of OPFRs in the sewage sludge.

| OPFRs | Sewage Sludge | | | |
| --- | --- | --- | --- | --- |
| | Min | Max | Average | Freq. |
| DPHP | 0.42 | 1.34 | 0.88 | 100 |
| DBP | 0.51 | 4.46 | 1.53 | 100 |
| BEHP | 279 | 655 | 449 | 100 |
| TCEP | 16.3 | 68.5 | 35.7 | 100 |
| TPP | 0.55 | 1.48 | 1.19 | 100 |
| TCPP | 36.1 | 2178 | 1251 | 100 |
| TDCPP | ND | 40.3 | 15.3 | 60 |
| TPHP | 1.09 | 36.7 | 16.7 | 100 |
| TDBPP | ND | 100 | 29.2 | 40 |
| TMPP | 13.3 | 37.2 | 24.8 | 100 |
| TEHP | 17.1 | 59.8 | 32.4 | 100 |

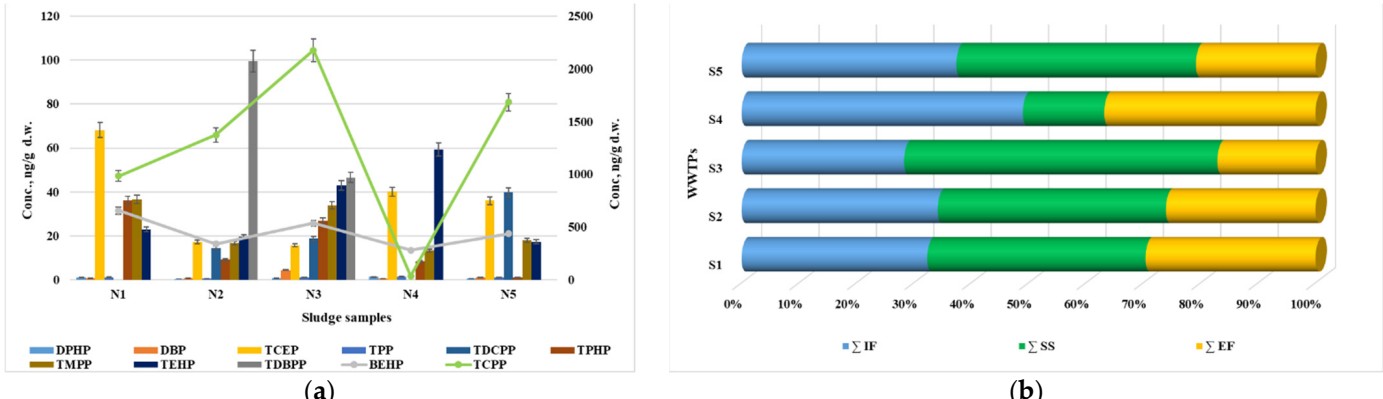

**Figure 4.** Composition of OPFRs in sewage sludge samples (**a**), and the OPFR distribution between the aqueous and solid components of WWTPs (**b**).

Figure 4b illustrates the distribution of the OPFRs among the influent, sludge, and effluent samples. Despite the low removal efficiencies, the results indicate that a substantial portion of the compounds was adsorbed onto the solid particles of the sludge. This can be attributed to the water–octanol partition coefficients (log Kow) and the soil organic carbon–water partitioning coefficient (log Koc), as shown in Table S2.

The results presented in Table 4 indicate the total sums of the OPFR compounds determined at the treatment plants, specifically in the influent, effluent, and sewage sludge samples. It is mentioned that no significant differences were observed after the treatment processes when comparing the influent and effluent sums of the OPFRs. The influent sums of the OPFRs ranged from 1507 to 2004 ng/L, while the effluent sums ranged from 909 to 1406 ng/L. These findings suggest that the treatment processes were not effective in significantly reducing the total concentrations of the OPFRs in the wastewater. This indicates potential challenges in removing these compounds using the current treatment methods employed.

**Table 4.** Sum of OPFRs quantified for each (RSD < 5%, *n* = 3).

| WWTPs | ∑OPFRs | | |
|---|---|---|---|
| | IF (ng/L) | SS (ng/g d.w.) | EF (ng/L) |
| S1 | 1537 ± 77 | 1807 ± 85 | 1406 ± 59 |
| S2 | 1627 ± 81 | 1893 ± 89 | 1241 ± 52 |
| S3 | 1507 ± 75 | 2905 ± 137 | 909 ± 38 |
| S4 | 1518 ± 76 | 439 ± 21 | 1139 ± 48 |
| S5 | 2004 ± 100 | 2235 ± 105 | 1115 ± 47 |

One notable observation from the results is the significant levels of the OPFRs detected in the sewage sludge samples. The sums of the OPFRs in the sewage sludge ranged from 439 to 2905 ng/g d.w. These values suggest a significant accumulation of the target compounds on the solid particles present in the sludge. This accumulation has potential environmental repercussions, particularly if the sewage sludge is utilized in agricultural practices. If the sewage sludge containing significant amounts of OPFRs is used as a fertilizer in agriculture, there is a risk of releasing these compounds into the environment. This can further contribute to the contamination of soil, water bodies, and potentially impact ecosystems.

Regarding the differences within the same system, the results highlight that there are minimal differences between the concentrations of the OPFRs in the influents (incoming wastewater) and effluents (treated wastewater) within the same station. This observation is further supported by the low removal yields calculated for each treatment station, suggesting that the treatment processes employed are not effectively removing OPFRs. Additionally, it is noted that the sludge concentrations of the OPFRs are similar across all the treatment styles. This implies that the sludge retains a consistent level of OPFR concentration regardless of the specific treatment method used. However, an exception is observed in station S4, where the OPFR concentration in the sludge is approximately 4.5 times lower compared to the other stations. This indicates that the treatment process employed at station S4 is more successful in removing OPFRs, specifically in this case. These findings indicate that the current treatment methods utilized in the stations are generally ineffective in removing OPFRs from wastewater, with the exception of station S4. This suggests that further investigation and improvements are required to enhance the overall efficiency of OPFR removal in the treatment stations.

To mitigate these risks, it is crucial to ensure the proper management and disposal of sewage sludge. Adequate treatment and disposal methods should be employed to minimize the release of OPFRs and other pollutants into the environment. Additionally, further research is needed to develop more effective treatment processes to remove OPFRs from wastewater and minimize their accumulation in sewage sludge. Overall, the findings give insight into the presence and behavior of OPFRs in wastewater treatment plants, highlighting the need for improved treatment strategies and the proper management of sewage sludge to prevent environmental contamination and promote sustainable practices.

### 3.2. Occurrence of OPFRs in Surface Waters

Similar to the previous findings, TCPP was also identified as the main contaminant in the surface waters (Table 5). The concentrations of TCPP ranged from 283 ng/L to 933 ng/L for the samples collected upstream of the WWTPs and from 670 ng/L to 1601 ng/L downstream of the WWTPs. These concentrations are comparable to those reported in the Danube River (26–603 ng/L) [53], as well as in Australia (up to 891 ng/L) [54] and China (up to 840 ng/L) [55]. However, they are lower than the concentrations reported in Spain (ND-1800 ng/L) [56], Canada (290–2010 ng/L) [57], and Korea (ND–5102 ng/L) [58], but higher than those observed in Germany (40–250 ng/L) [59], the UK (26–133 ng/L) [60], and France (<98.2 ng/L) [61].

**Table 5.** Minimum, maximum, and average concentrations (ng/L) and determination frequencies (%) of OPFRs in the surface waters.

| OPFRs | Upstream | | | | Downstream | | | |
|---|---|---|---|---|---|---|---|---|
| | **Min** | **Max** | **Average** | **Freq.** | **Min** | **Max** | **Average** | **Freq.** |
| DPHP | 1.02 | 7.14 | 2.98 | 100 | 0.78 | 5.53 | 2.64 | 100 |
| DBP | 2.08 | 28.3 | 15.3 | 100 | <LOQ | 24.8 | 9.18 | 100 |
| BEHP | 0.81 | 32.6 | 14.7 | 100 | 11.5 | 30.6 | 17.3 | 100 |
| TCEP | 7.73 | 22.1 | 13.9 | 100 | 16.2 | 141 | 52.8 | 100 |
| TPP | 0.54 | 1.2.5 | 0.92 | 100 | 0.53 | 1.04 | 0.73 | 100 |
| TCPP | 283 | 933 | 579 | 100 | 670 | 1601 | 1036 | 100 |
| TDCPP | 4.33 | 11.8 | 7.47 | 100 | 6.14 | 61.3 | 24.5 | 100 |
| TPHP | 3.76 | 7.43 | 5.14 | 100 | 3.67 | 9.44 | 6.06 | 100 |
| TDBPP | ND | 45.4 | 11.2 | 40 | ND | 37.8 | 15.4 | 60 |
| TMPP | 4.22 | 19.8 | 11.8 | 100 | 2.98 | 16.1 | 7.11 | 100 |
| TEHP | 17.1 | 31.5 | 23.4 | 100 | 8.13 | 13.4 | 10.1 | 100 |

Average concentrations below 100 ng/L were determined for DBP, BEHP, TCEP, TDCPP, TDBPP, TMPP, and TEHP, with maximum values ranging from 11.2 ng/L to 23.4 ng/L upstream and 9.18 ng/L to 52.8 ng/L downstream (Figure 5).

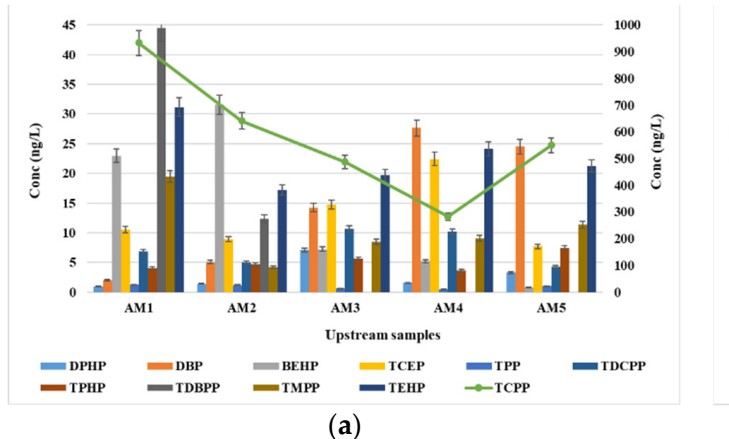

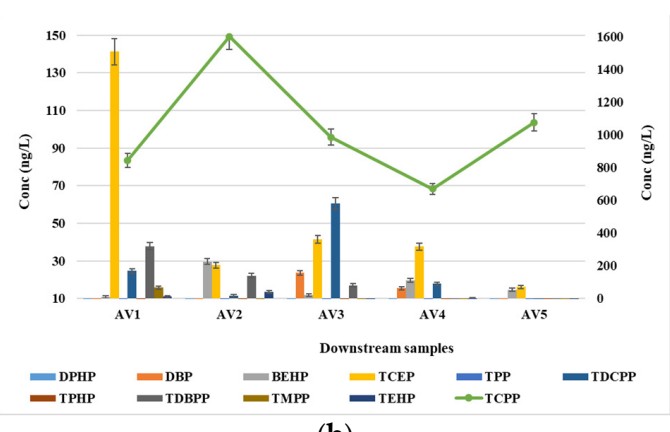

(**a**)        (**b**)

**Figure 5.** Composition of OPFRs in samples collected upstream (**a**) and downstream (**b**) of the WWTPs.

The presence of TCEP in the wastewater, coupled with its inefficient removal during the treatment processes has led to the increase in the concentration in the samples collected downstream of the WWTPs. The obtained values are of a similar order of magnitude to those reported in studies conducted in Australia [55] and Canada [58], lower compared to the UK [60], Spain [56], Korea [58], and China [55], but higher than those reported in France [61], Italy [62], and Sweden [63].

*3.3. Transfer of OPFRs from WWTPs to the Aquatic Environment*

The presence of OPFRs in all the analyzed matrices indicates the widespread occurrence of these compounds in the Romanian environment (Figure 6a). The surface waters are already contaminated with a significant amount of OPFR compounds even upstream of the WWTPs. The contribution of the WWTPs' effluents to the overall OPFR load in the receiving rivers is, in some cases, substantial, suggesting that the inefficiency of removing these compounds in wastewater treatment steps is a major contributing factor.

In both the wastewater and surface water, TCPP was the predominant compound (Figure 6b). Despite its relatively low water–octanol partition coefficient (2.59), showing a greater affinity for the aqueous samples than for solid ones, TCPP was also frequently detected in the solid samples. In sludge samples, high levels of BEHP were also observed.

The presence of BEHP in the solid matrix can be attributed to its tendency to adsorb onto solid particles, as evidenced by its high log Kow (6.07) and log Koc (4.23) values.

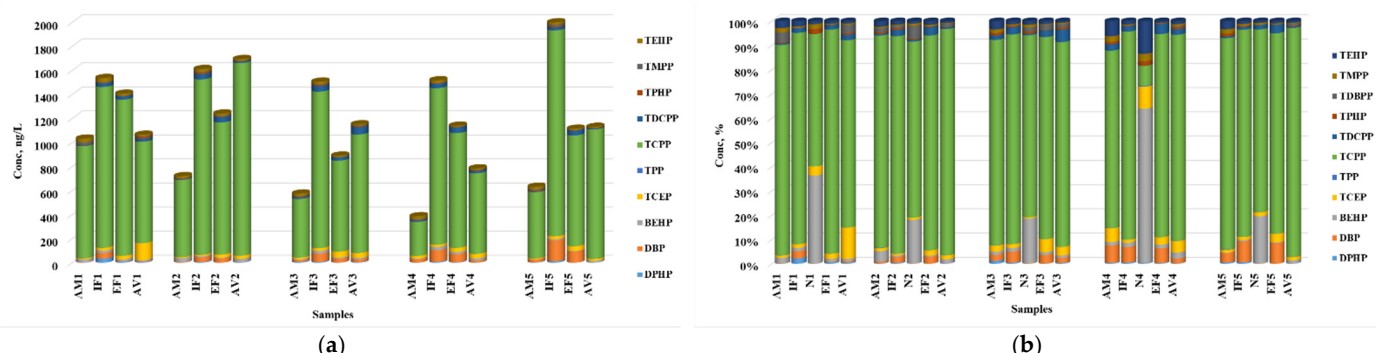

(**a**)            (**b**)

**Figure 6.** OPFR occurrence in wastewater and surface water samples (**a**), and percentage distribution of OPFRs in surface water, waste water, and sludge samples (**b**).

The transfer mechanisms of the OPFRs from the WWTPs to the surface waters involve various processes, including release, transport, and fate within the wastewater treatment process. OPFRs can enter WWTPs through multiple pathways. Industrial discharges, domestic sources such as households and hospitals, as well as agricultural runoff contribute to the input of OPFRs into the wastewater system. Once within the WWTP, OPFRs go through different treatment processes. In the primary treatment phase, physical separation processes such as sedimentation and screening are employed to remove solid materials, including larger debris and suspended solids. Some OPFRs may associate with these solid particles and settle during this phase, leading to their removal from the liquid phase. However, the effectiveness of removal during this stage varies depending on the characteristics of the compounds and the treatment technologies utilized. Secondary treatment processes aim to eliminate organic matter and nutrients from the wastewater. Common methods, like activated sludge or biological treatment, involve microorganisms breaking down organic compounds in the wastewater, including OPFRs. The removal efficiency of OPFRs through biological treatment differs based on their biodegradability and the specific operational conditions of the WWTP. Sorption onto the sludge generated during treatment is another possible fate for OPFRs. They may adsorb to organic matter present in the wastewater or the sludge, reducing their release into the liquid phase and subsequently limiting their transport to receiving rivers. Furthermore, OPFRs can undergo transformation reactions within the WWTP, leading to the formation of degradation products with potentially different properties.

The effluents are discharged from WWTPs into receiving rivers. The composition of OPFRs in the effluent is influenced by their removal efficiency during the treatment processes. The effluent can transport OPFRs and their degradation products to surface waters, potentially impacting aquatic ecosystems. The actual transport mechanisms and fate of OPFRs in surface waters are dependent on their physicochemical properties, interactions with sediment or organic matter, hydrological conditions, and environmental processes such as photolysis and biodegradation. Exploring the specific mechanisms involved in OPFRs' transfer from WWTPs to surface waters in Romania would require further research and investigation. Factors like WWTP infrastructure, operational conditions, influent OPFR concentration and composition, and the prevalence of specific treatment technologies can all affect the overall fate and transport of OPFRs within the wastewater treatment process.

### 3.4. Environmental Risk Assessment

To assess the potential risk of OPFR compounds in the surface waters of Romania, the estimated risk quotients (RQs) were calculated considering the high concentrations detected in the samples collected both upstream and downstream of the WWTPs. Given that these compounds are known to be toxic to aquatic organisms and have endocrine-disrupting

properties, it is important to evaluate the potential risk factors they pose to the aquatic environment [56]. Equations (4) and (5) were used in the evaluation of the potential risks on the environment from OPFRs based on toxicological studies reported in the literature (Table S11) [40,60]. These equations are commonly used in environmental risk assessments to estimate the predicted no-effect concentration (PNEC) and calculate the risk quotient (RQ). Equation (4) calculates the PNEC by dividing the no-effect concentration (NOEC) or the median lethal concentration ($LC_{50}$) or median effective concentration ($EC_{50}$) by an assessment factor (AF). The AF accounts for uncertainties and factors such as differences in the sensitivity between the tested organisms and the environment. In this case, an AF value determined in accordance with the technical guidance document (TGD) on risk assessment from the European Commission is used. Different AF values are applied depending on the type of toxicity information available. For example, an AF value of 1000 is applied for $LC_{50}/EC_{50}$ (acute toxicity), 100 for long-term NOEC (chronic toxicity), 50 for two long-term NOECs, and 10 for three long-term NOECs in species representing different trophic levels.

In this paper, the environmental risk for aquatic species was evaluated for several determined OPFRs, including TCEP, TDCPP, TCPP, TPHP, TEHP, TMPP, TPP, and TDBPP in surface water samples. The results revealed that, for most of the compounds, the risk quotients were below 0.1, indicating that the majority of OPFRs did not pose a risk to aquatic organisms. However, it was observed that the downstream concentrations of TDCPP and TCPP in the WWTP effluents present a moderate risk for certain fish species. Additionally, TMPP both upstream and downstream of the WWTPs showed a moderate risk for algae. These findings are consistent with previous studies where mixtures of OPFRs were frequently detected in surface waters, posing a low to moderate risk to biota in aquatic ecosystems [64]. Similarly, studies worldwide have reported a moderate risk to aquatic organisms from TCPP [27,65].

Although the risk assessment for the targeted OPFRs was generally low, further monitoring research is warranted, as several studies have demonstrated that many halogenated OPFRs are not effectively removed during wastewater treatment [14]. Moreover, recent papers have communicated the occurrence of OPFRs in biota [27,66], suggesting their potential bioaccumulation in various organisms. Therefore, future surveys investigating the bioaccumulation and biomagnification of OPFRs in aquatic organisms are crucial for a comprehensive risk evaluation.

The potential long-term effects of OPFRs and their ability to bioaccumulate in aquatic organisms are significant concerns for the environment. OPFRs have been found to exhibit various toxic effects on aquatic organisms, including fish, invertebrates, and algae. These compounds can interfere with hormonal systems, disrupt growth and development, impair reproductive success, and cause adverse effects on overall organism health [30,32]. These impacts can have cascading effects on aquatic ecosystems, affecting population dynamics, food webs, and ecosystem functioning.

One key concern is the potential for OPFRs to bioaccumulate in aquatic organisms. Bioaccumulation refers to the gradual accumulation and persistence of a substance in an organism's tissues over time. OPFRs have properties that make them prone to bioaccumulation. They have a low water solubility and high affinity for organic matter, allowing them to partition into lipid-rich tissues of organisms. The bioaccumulation of OPFRs can occur through various pathways. Aquatic organisms can directly take up OPFRs from contaminated water or sediments, and they can also ingest OPFRs indirectly through the consumption of contaminated prey. Once inside an organism, OPFRs can be absorbed, metabolized, or eliminated. However, if an organism's uptake rate exceeds its elimination rate, bioaccumulation can occur. The extent of the bioaccumulation depends on factors such as the specific OPFR compound, its concentration in the environment, the lipid content of the organism, and the duration of exposure. The potential for OPFRs to bioaccumulate raises concerns because it can lead to increasing concentrations of these compounds at higher trophic levels in the food chain. Predatory organisms at the top of the food chain, such as large fish or marine mammals, can accumulate higher concentrations of OPFRs

than lower trophic level organisms. This biomagnification of OPFRs poses risks because organisms at higher trophic levels may experience more pronounced adverse effects due to higher exposure levels [27,28]. Additionally, if humans consume contaminated seafood, they may be exposed to OPFRs through the food chain.

It is important to note that the bioaccumulation potential of OPFRs can vary among different compounds. Each OPFR compound possesses unique properties that may influence its ability to bioaccumulate. Some compounds may have a relatively low bioaccumulation potential, while others may exhibit higher bioaccumulation rates.

To fully comprehend the long-term effects and bioaccumulation potential of OPFRs, further research is necessary. This entails studying the persistence of OPFRs in the environment, monitoring their levels in aquatic organisms, and conducting ecotoxicological studies to evaluate the chronic effects of OPFR exposure on aquatic ecosystems. Such research is invaluable for developing comprehensive risk assessments and regulatory measures to mitigate the environmental and health risks associated with OPFRs.

## 4. Possible Mitigation or Control Strategies of OPFR Contamination

The recommendations provided above highlight different strategies that can be employed to mitigate or control the OPFR contamination in aquatic environments. These strategies vary from regulatory measures to wastewater treatment, source control, public awareness, and research and innovation. Regulatory measures play a crucial role in managing the use and release of OPFRs. Governments can enact policies and regulations that restrict or ban the use of certain OPFRs with a high toxicity and bioaccumulation potential. By implementing such measures, the input of these harmful compounds into the environment can be minimized, preventing their negative impacts on aquatic organisms. Enhanced wastewater treatment is another effective strategy. Upgrading wastewater treatment plants with advanced technologies can help remove OPFRs from contaminated water before it is released into water bodies. Techniques such as activated carbon adsorption, ozonation, or membrane filtration can target the removal of these compounds, reducing their concentration in aquatic environments.

Source control is also important in preventing OPFR contamination. Industries that use OPFRs in their products can explore alternative flame retardants with a lower toxicity and bioaccumulation potential. By shifting to safer alternatives, the input of OPFRs into the environment can be reduced. Public awareness and education are crucial for driving change. By increasing awareness among consumers, industry members, and policymakers about the risks associated with OPFRs, demand for safer alternatives can be created. Labeling requirements and certification programs can also help consumers make informed choices while purchasing products, promoting the use of safer flame retardants. Research and innovation are essential for finding alternative flame retardants that are effective, safe, and environmentally friendly. By investing in research and development, we can identify and promote the use of flame retardants that are less hazardous and have lower persistence in the environment. Collaboration and international standards are also important in addressing OPFR contamination. By working together, countries can develop consistent and harmonized standards for flame retardants, ensuring the adoption of safer alternatives globally.

It is important to note that implementing these strategies requires a multidisciplinary approach involving governments, industries, academia, and the public. Cooperation and collaboration among these stakeholders are necessary to effectively mitigate and control OPFR contamination and protect the health of aquatic ecosystems and human populations. The ongoing monitoring and evaluation of the effectiveness of these strategies are also essential to make necessary adjustments and improvements.

## 5. Conclusions

In this research, the concentration, distribution profile, and potential environmental risk of OPFRs were determined and analyzed in five WWTPs and natural receivers in

Romania. All 11 OPFRs were detected in the samples. Among them, TCPP was the most prevalent contaminant, with a concentration up to 1422 ng/L in wastewater, 1851 ng/g d.w. in sewage sludge, and up to 1036 ng/L in surface waters. The WWTPs showed poor removal efficiencies for all evaluated OPFRs. Daily mass loading and daily mass emission calculations revealed high values, particularly for TCPP. The distribution profile of the OPFRs was mainly influenced by their physicochemical characteristics, such as log kow and log Koc. The environmental risk and health risk assessment indicated that some OPFRs may pose a moderate risk to aquatic species, while the majority of the OPFRs were not found to be a threat to aquatic biota. This research provides valuable insights for the development and implementation of control strategies to mitigate OPFR contamination in relevant environmental compartments in Romania and also supports international pollutant research efforts.

**Supplementary Materials:** The following supporting information can be downloaded at https://www.mdpi.com/article/10.3390/jox14010003/s1, Tabel S1: Collection points and sample codification; Table S2: Physical-chemical properties of the OPFR compounds; Table S3: Condition of instrumental analysis for OPFR compounds; Table S4: Gradient elution program; Table S5: Acquisition time-segments; Table S6: MRM conditions of OPFR (Q—quantifier, q—qualifier); Figure S1: MRM chromatogram registered for a standard mixture of 1 µg/L; Tabel S7: RSD values for intra-day and inter-day precision obtained for a spike concentration of 1 µg/L in each environmental matrix; Table S8: Recovery values obtained for all types of matrices (spike concentration = 1 µg/L, *n* = 3); Table S9: Matrix effect (ME) values determined for the targeted analytes in all environmental samples (post-extraction spike = 1 µg/L, *n* = 3); Table S10: Instrumental limit of quantitation (IOQ), and method limit of detection and quantitation obtained for the targeted compounds; Table S11: Acute toxicities (LC50 and EC50) used to calculate risk quotients; Table S12: Environmental risk of OPFRs for aquatic organisms. References [67–70] are cited in the supplementary materials.

**Author Contributions:** Conceptualization, L.F.P. and F.L.C.; methodology, I.P. and F.P.; software, V.I.I.; validation, I.P., F.P. and M.N.; formal analysis, I.P., V.I.I. and F.P.; investigation, F.P. and F.L.C.; resources, L.F.P. and V.I.I.; data curation, V.I.I.; writing—original draft preparation, I.P. and F.P.; writing—review and editing, L.F.P. and F.L.C.; visualization, V.I.I. and M.N.; supervision, L.F.P. and F.L.C.; project administration, F.L.C.; funding acquisition, V.I.I. and M.N. All authors have read and agreed to the published version of the manuscript.

**Funding:** This research was funded by the Romanian Ministry of Research, Innovation and Digitalization grant number 3N/2022, project code PN 23 22 01 01.

**Institutional Review Board Statement:** Not applicable.

**Informed Consent Statement:** Not applicable.

**Data Availability Statement:** No new data were created or analyzed in this study. Data sharing is not applicable to this article.

**Acknowledgments:** This work was carried out through the "Nucleu" Program within the National Research Development and Innovation Plan 2022–2027.

**Conflicts of Interest:** The authors declare no conflict of interest.

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
