# Peer review of "An Initial Survey on Occurrence, Fate, and Environmental Risk Assessment of Organophosphate Flame Retardants in Romanian Waterways"

_jox, doi:10.3390/jox14010003_

Round 1

Reviewer 1 Report

Comments and Suggestions for Authors

General Comments:
- The manuscript is somewhat well written, providing a clear overview of the research objectives, methodology, and key results. Just some comments and suggestions for the further improvement of the paper.

Abstract:
- Provide a brief explanation of the potential sources of OPFRs.

Introduction:
- Include information on the characteristics of Romanian freshwater ecosystems, the extent of pollution, and the unique challenges faced by the region.
- Elaborate on the specific methodologies used in the study by including a brief discussion on the chosen monitoring techniques and their effectiveness.

Methodology:
- Section 2.2. Sample collection: Indicate when each of the samples (influent, effluent, and sludge) was collected from five WWTPs in Romania. All in all, how many samples were collected? How long is the sampling?

Results:
- This Section 3 should be written as "Results and Discussion".
- Section 3.1.1. Explain further why TDBPP had a determination frequency of only 20%.
- Add a comprehensive discussion on the mechanisms of OPFRs transfer from WWTPs to surface waters.  How these compounds are released and transported through the wastewater treatment process to the receiving rivers?
- Add discussion on potential long-term effects or the possibility of OPFRs bioaccumulation in aquatic organisms.

Conclusion:
- Add recommendations on the possible mitigation or control strategies of OPFR contamination.

Comments on the Quality of English Language

Minor editing of English language required

Author Response

Dear reviewer,

We express our sincere gratitude for your time and effort in reviewing our paper. Your valuable observations and suggestions have greatly contributed to the improvement of our manuscript.

We truly appreciate your thorough review, which evidenced your expertise in the field. Your insightful comments have not only helped us address the shortcomings and refine our research but have also added significant value to the paper.

Your feedback has been particularly invaluable. We have carefully considered your suggestions and have made the necessary revisions to enhance the clarity, accuracy, and overall quality of the manuscript.

Once again, we would like to extend our heartfelt thanks for your time, dedication, and expertise in reviewing our work. We are incredibly grateful for your contribution, and we believe that your feedback has significantly strengthened our paper. We acknowledge the significant impact that your critical input has had on the final version of the manuscript.

Thank you once again for your invaluable support—we deeply appreciate your time, effort, and dedication.

Please find attached the responses to your suggestion

Reviewer 2 Report

Comments and Suggestions for Authors

The manuscript "Occurrence, fate and environmental risk assessment of organophosphate flame retardants: first survey in WWTPs and rivers from Romania" is a clear, succinct, and well written article. As far as I am concerned, this manuscript could be accepted in its present form.

Author Response

(The authors gave the same response as above.)
